# Psychometric Properties of the Japanese Version of the Parental Acceptance and Action Questionnaire in Parents with Infants and Toddlers

**DOI:** 10.3390/ijerph20095674

**Published:** 2023-04-28

**Authors:** Junko Okajima, Isa Okajima

**Affiliations:** 1Department of Psychology, Rikkyo University, Saitama 352-8558, Japan; 2Department of Rehabilitation, University of Tokyo Health Sciences, Tokyo 206-0033, Japan; 3Department of Psychological Counseling, Faculty of Humanities, Tokyo Kasei University, Tokyo 173-8602, Japan

**Keywords:** parents, acceptance and commitment therapy, psychological flexibility

## Abstract

We aimed to examine the reliability and validity of the Parental Acceptance and Action Questionnaire-Japanese version (PAAQ-J). We considered a total of 2000 mothers with infants and toddlers aged 0–3 years and evaluated their scores on the PAAQ-J Acceptance and Action Questionnaire-II (AAQ-II) and Hospital Anxiety and Depression Scale (HADS). We conducted an exploratory factor analysis, creating a PAAQ-J with 12 items and three factors (α = 0.80): Inaction-Behavior, Inaction-Cognition, and Unwillingness, with α of 0.84, 0.72 and 0.68, respectively. The test-retest reliability examination results showed that the interclass correlation coefficient was 0.49, with 95% CI between 0.44 and 0.54. The correlation coefficient of PAAQ-J was 0.57, 0.32, and 0.33 with AAQ-II, and HADS-depression and HADS-anxiety, respectively. PAAQ-J’s validity to adequately evaluate an individual’s avoidance of experiences regarding childcare and their psychological flexibility was proven. Since the original PAAQ was for 6–18-year-old children with anxiety symptoms, it is necessary to examine its reliability and validity not only for infants and toddlers, but also for parents of older children and adolescents in the future.

## 1. Introduction

In 2021, the number of consultations and responses to child abuse in Japan was 207,660, 101.3% of the previous year (an increase of 2616 cases), the highest number ever recorded [1]. Furthermore, it is known that 77.2% of the fatal accidents due to abuse between 2003 and 2009 involved children aged 3 years or younger, and that biological mothers accounted for the largest percentage of perpetrators at 55.1%. Childcare stress is considered to be particularly high for mothers caring for children from infancy to 3 years of age. During infancy, infants undergo dramatic growth and change, and the stressors felt by mothers also change significantly [2]. For example, in infants, the main stressors are irregular sleep, night crying, and feeding, while around one year of age, problems related to weaning, walking, and language development are added [3]. Then, around the age of three, they enter a period of rebellion, and emotional and personality problems such as not listening to their parents and fighting in interpersonal relationships are added [3]. Inappropriate emotional regulation by mothers has been shown to have a significant impact on children’s emotional regulation skills and mental health, including children’s modeling of parental emotional regulation skills and instability in children’s attachment systems [4]. In light of these findings, some kind of parental support for mothers raising infants to 3-year-old toddlers is desirable. In recent years, much attention has been paid to attempts to adapt Acceptance and Commitment Therapy (ACT) in parent support [5].

ACT is part of traditional Cognitive Behavioral Therapy (CBT), but specifically focuses on the function and context of psychological phenomena rather than the content and form of thoughts, feelings, and sensations [6]. ACT aims to further psychological flexibility by promoting openness and awareness and enabling flexible conduction of actions and adoption of behaviors that are backed up by an individual’s values [7]. Psychological flexibility is enhanced through the six core pathological processes: mindfulness, acceptance, defusion, self-as-context, committed action, and values [8].

ACT can be useful for caregivers who manage various difficulties by targeting experiential avoidance and cognitive fusion [9]. Previous studies have shown that caregivers who rely on experiential avoidance tend to control or suppress psychological pain related to a child’s difficulties, without taking appropriate actions backed by adequate behaviors [9]. Unlike conventional cognitive behavioral approaches, ACT specifically focuses on values. Engaging in actions and behaviors without avoiding unpleasant experiences enables the adoption of long-term behaviors [10]. A systematic review of ACT based on parental support concludes that the approach is a promising intervention that helps parents manage the stress and difficulties associated with autism [11], pediatric diseases [12,13], and children’s chronic pain [14,15]. In this context, ACT-based parental support and assessment tools for parents are actively being developed [16,17].

Parental psychological flexibility is closely related to child-rearing and parenting [18], and several studies have reported on psychological flexibility, parental mental health, and its impact on children. For example, Brassella and his colleagues [19] conducted a study on psychological flexibility in parents with children who are in early childhood (3–7 years), middle childhood (8–12 years), and adolescence (13–17 years). They found that the higher the parental-specific psychological flexibility, the lower the level of internalization and externalization problems in adolescents through adaptive child-rearing practices. This was revealed to be consistently relevant in any age group of children. Next, a survey by Fonseca et al. [20] revealed that child-rearing stress directly affects the child-rearing style of mothers with children between the ages of 2 and 12 years, and indirectly affects the child-rearing style through the psychological flexibility of mothers. These are immutable among mothers of different age groups, and it became evident that the psychological flexibility of mothers was positively associated with the use of maladaptive child-rearing styles (authoritarian, tolerant) and negatively associated with the use of authoritative child-rearing styles. From these, it is clear that there is a close relationship between parents’ psychological flexibility and parenting behavior, and that it has an impact on children. Furthermore, an accurate measurement of parental psychological flexibility for mothers raising infants to toddlers is thought to be useful for the development of parental support through ACT.

Some measures of parental psychological flexibility have been developed in the literature, and Japanese versions of the Parent Acceptance Questionnaire (6-PAQ) and the Parent Acceptance and Behavior Questionnaire (PAAQ) have been developed. The 6-PAQ is designed such that all of the six processes of psychological flexibility, namely, acceptance, defusion, self-as-context, being present, values, and committed action, can be evaluated. Since it is standardized for parents with children who are 3 to 12 years old, there is an item, “When my child misbehaves” that is tailored to the behavior of the children. Therefore, it is thought that adapting it to parents with infants and toddlers is difficult. We also describe the target ages of the Japanese version.

The PAAQ was created based on the AAQ, with appropriate changes to the wording of each question to address respondents’ children and their individual parenting behaviors towards those children [21]. For example, the original item, “If I could magically remove all of the painful experiences I’ve had in my life, I would do so” was modified to “If I could magically remove all of the painful experiences my child has in his or her life, I would do so”. A factor analysis of PAAQ confirmed that it is a two-factor structure: parents who do not want to see their children experience negative emotions (the Unwillingness Subscale), and parents who were unable to effectively manage their responses to their children’s emotions (the Inaction Subscale).

The original PAAQ is a scale specifically designed to assess caregivers’ experiential avoidance of parenting [21]; the 15-item, seven-point self-administered scale was developed to measure psychological flexibility, the Acceptance and Action Questionnaire (AAQ; [22]), which is based on the AAQ. The target population for standardization of this scale included 154 children aged 6 to 18 years (90 females and 64 males) diagnosed with anxiety disorders and their parents (148 mothers and 119 fathers). Factor analysis of the PAAQ resulted in a two-factor solution with factors labeled Inaction and Unwillingness. The temporal stability of the PAAQ was moderate (r = 0.68–0.74), and the internal consistency among the PAAQ subscales was also reliable, α = 0.64–0.65. Because this study wanted to measure experiential avoidance among parents raising infants and toddlers, the PAAQ was used to examine reliability and validity. The Japanese version of the PAAQ-J (PAAQ-J) was developed by Mizusaki and Sato [23], who attempted to standardize it for parents with teenage children (*n* = 47).

Thus, the item contents of PAAQ are more likely to be better rated than 6-PAQ to measure the psychological flexibility of parents that are raising infants and toddlers. Therefore, we aimed to examine reliability and validity of the PAAQ-J in parents with infants and toddlers.

## 2. Materials and Methods

### 2.1. Participants

Data were collected in December 2020. Participants were recruited by Rakuten Research, Inc., an online marketing research company that possesses the contact details of approximately 2.3 million Japanese survey respondents. Randomly selected individuals from Japan, stratified by gender and age, were sent an e-mail containing a link to an online questionnaire.

The participants were 2000 mothers of children aged 0–3 (500 mothers of 0-year-olds, 500 mothers of 1-year-olds, 500 mothers of 2-year-olds, and 500 mothers of 3-year-olds); the children’s mean age was 1.57 ± 0.74 years; the mothers’ was 33.58 ± 4.7 years. The inclusion criterion was mothers of children aged 0–3 years. There were no exclusion criteria. Mothers who had multiple children in the targeted age group were asked to complete the questionnaire considering only one child. Of these, 1000 participants (223 one-year-olds, 234 two-year-olds, 291 three-year-olds, and 252 four-year-olds) engaged in another survey fourteen months later to examine test-retest reliability.

### 2.2. Measures

Demographic information. Participants were asked to provide information regarding their age, employment status, number of births, mental health, and specific physical health issues (premenstrual syndrome [PMS], premenstrual dysphoric disorder [PMDD]). We also collected information regarding number of children, age of the target child, and child birth order.

The PAAQ-J is a 15-item self-reporting questionnaire and is evaluated on a seven-point scale (from 1 [Never True] to 7 [Always True]) [23]. The PAAQ-J was developed and structured by Mizusaki and Sato [23], however, reliability and validity were insufficiently examined due to the small sample size (*n* = 47). The higher the PAAQ-J score, the higher the tendency to avoid experiences. The questionnaire comprises two subscales, “Inaction” and “Unwillingness”. “Inaction” indicates a caregiver’s inability to functionally control their reactions to a child’s emotions, and “Unwillingness” indicates their inhibition to witness a child’s negative emotional experiences. The α-coefficients for “Inaction” and “Unwillingness” in the original PAAQ were 0.64 and 0.65, respectively; the total α-coefficient was 0.65. The PAAQ-J was reviewed by two clinical psychologists (JO, IO), who determined that the item content was adaptable to parents raising children in infancy.

Acceptance and Action Questionnaire-II (AAQ-II). The AAQ-II is a self-reporting questionnaire comprising seven items that evaluate important aspects of an adult’s avoidance of experiences, and psychological flexibility, on a seven-point scale (from 1 [Never True] to 7 [Always True]). The higher the score, the greater an individual’s tendency to avoid experiences [10]. The Japanese edition of AAQ-II, which has an α-coefficient of 0.88, representing high reliability and validity, was developed by Shima, et al. [24].

Hospital Anxiety and Depression Scale (HADS). The HADS is a 14-item scale that evaluates anxiety and depressive symptoms; it comprises seven items each for anxiety and depressive symptoms [25]. The participants are asked to answer questions about each symptom on a scale of 1–4. The higher the score, the stronger the symptom. The Japanese edition of HADS was developed by Hatta, et al. [26]. Cronbach’s α was 0.80 for anxiety symptoms, and between 0.59 and 0.61 for depressive symptoms.

### 2.3. Method of Analysis

This study examined the PAAQ-J’s reliability and validity according to COSMIN’s framework [27] (see also [28]).

First, to verify structural validity, we conducted confirmatory factor analysis (CFA), and assessed whether a two-factor structure could be hypothesized for PAAQ-J, as in the original PAAQ [21]. Whenever a two-factor structure was invalid, we used exploratory factor analysis, employing maximum likelihood promax rotation to investigate a new factor structure. The final extracted factors were subjected to CFA to ascertain the degree of fit.

Further, to examine the reliability of PAAQ-J, Cronbach’s α coefficients were calculated for each of the factors. We then calculated the intraclass correlation coefficient (ICC) for each factor to examine the test-retest reliability of the PAAQ-J.

We also conducted a correlation analysis with AAQ-II to ascertain the PAAQ-J’s criterion validity. We further conducted a correlation analysis between PAAQ-J and HADS-depression and HADS-anxiety to investigate construct validity. We used SPSS Statistics ver. 27 (IBM Corp., Armonk, NY, USA) and SPSS AMOS ver. 27 (IBM Corp., Armonk, NY, USA) for statistical analyses.

## 3. Results

### 3.1. Demographic Data

Table 1 shows the demographic data of the participants. The employment status at the baseline contained full-time workers (28%), part-time workers (8.1%), maternity and childcare leaves (21.4%), and homemakers (41.4%). The child’s birth order was 1.15 ± 0.74; number of births 2.66 ± 0.80; psychiatric disorders (2.5%), PMS (12.7%), and PMDD (2.7%).

### 3.2. Structural Validity

We conducted a CFA that hypothesized a two-factor structure similar to the original PAAQ [21] and found that the model’s goodness of fit was poor (CMIN/DF = 41.363; CFI = 0.586; GFI = 0.754; AGFI = 0.668; RMSEA = 0.142; Figure 1).

An exploratory factor analysis was subsequently performed. Kaiser–Meyer–Olkin (KMO, 0.831) and Bartlett’s sphericity tests were significant (*χ*^2^ = 105, 8755.563; *p* < 0.001), it was deemed appropriate to perform factor analysis. A three-factor structure was determined based on the shape of the scree plot. We then conducted a factor analysis, employing maximum likelihood promax rotation. Double-load items (Items 4 and 6) and items which had their factor loading below 0.30 (Item 5), were eliminated before the analysis was re-conducted. The results showed that Factor 1 comprised items 1, 7, 10, and 12; Factor 2 comprised items 2, 3, 14, and 15; and Factor 3 comprised Items 8, 9, 11, and 13 (Table 2). Factor 1 comprised items related to the behavioral aspects included in “Inaction”, which is a subscale in the original PAAQ. Therefore, we named it “Inaction-Behavior (Inaction-B)”. Factor 2 comprised items related to cognitive aspects included in “Inaction”, another subscale in the original PAAQ. Hence, we named it “Inaction-Cognition (Inaction-C)”. Factor 3 comprised items similar to “Unwillingness”, which is another subscale in the original PAAQ. We, therefore, named it “Unwillingness”. These factors explained 59.8% of the total variance. The extracted factors were subjected to CFA. A model assuming a higher-order factor “Inaction” for “Inaction-B” and “Inaction-C” was examined. Modification indices were calculated, and covariation was assumed between errors for those with χ-square improvements greater than 0.3, resulting in a good fit (CMIN/DF = 9.261; CFI = 0.946; GFI = 0.936; AGFI = 0.940; RMSEA = 0.064; Figure 2).

### 3.3. Reliability

The internal consistency of Inaction-B, Inaction-C and “Unwillingness” was 0.84, 0.72, and 0.68 respectively; the overall consistency (α) of PAAQ-J was 0.80. The results of the test-retest reliability examination showed that Inaction-B, Inaction-C, and “Unwillingness” had an ICC [95% CI] of 0.47 [0.42, 0.52], 0.41 [0.36, 0.46], and 0.30 [0.24, 0.36], respectively; the overall ICC of the PAAQ-J was 0.49 [0.44, 0.54]. The results achieved by calculating other scales further indicated that AAQ-II, HADS-depression, and HADS-anxiety had an ICC of 0.59 [0.55, 0.63], 0.58 [0.54, 0.62], and 0.37 [0.32, 0.42], respectively. In the “Inaction” of the higher-order factor model, the internal consistency was 0.74, and test-retest reliability showed that “Inaction” had an ICC of 0.58 [0.54, 0.62].

### 3.4. Criterion and Construct Validity

We further conducted a correlation analysis to verify criterion validity. The results showed that the PAAQ-J had a medium-level positive correlation with AAQ-II (*r* = 0.57, *p* < 0.001) (Table 3). The correlation between the factors of AAQ-II and PAAQ-J (namely Inaction-B, Inaction-C, and “Unwillingness”) was *r* = 0.01 (n.s.), *r* = 0.59 (*p* < 0.001), and *r* = 0.35 (*p* < 0.001), respectively. We further conducted a correlation analysis between PAAQ-J and HADS-depression and HADS-anxiety to verify construct validity. The results indicated a significantly weak positive correlation between the scales mentioned above (*r* = 0.32, and *r* = 0.33, *p* < 0.001, respectively). The results of the correlation analysis conducted between the numerous factors of PAAQ-J and HADS-depression and HADS-anxiety were Inaction-B (*r* = 0.30, and *r* = 0.30, *p* < 0.001), Inaction-C (*r* = 0.19, and *r* = 0.27, *p* < 0.001), and “Unwillingness” (*r* = 0.02, n.s., and *r* = 0.09, *p* < 0.001) respectively. The results of the correlation analysis conducted between the various higher-order factor “Inaction”, HADS-depression and HADS-anxiety, and AAQ-II were *r* = 0.36, *r* = 0.34, and *r* = 0.48, *p* < 0.001, respectively.

## 4. Discussion

In this study, we aimed to examine whether the PAAQ is appropriate as a measure of psychological flexibility for parents that are raising infants to toddlers aged 0–3 years. By examining its validity and reliability we can measure the psychological flexibility of parents with infants and toddlers.

### 4.1. Structural Validity

A total of 154 children aged 6 to 18 years diagnosed with anxiety disorder and their parents participated in previous study of the original PAAQ. As a result of exploratory factor analysis, a scale consisting of two factors (“Inaction” and “Un-willingness”) and 19 items was developed [21]. In addition, for the PAAQ-J, 47 parents with teenage children participated in that study. The two-factor structure was not confirmed, and it was finally concluded that it was a one-factor structure. In our investigation, three items were removed as a result of an exploratory factor analysis, but out of the three items, Factor 1 and Factor 2 included the item “Inaction”, and Factor 3 included the item “Unwillingness”. The structures of the original PAAQ and PAAQ-J are similar. As “Inaction” was divided into behavior and cognition, the tendency to avoid experiences among Japanese who raise infants may differ in terms of behavioral and cognitive aspects. Furthermore, CFA was conducted assuming “Inaction” as a higher-order factor of “Inaction-B” and “Inaction-C”, and the goodness of fit was obtained. As an example of avoidance of experience in parenting, Coyne and Murrell [29] present the latter. “Trying not think about something when it ‘shows up’ in your thoughts”, and “Avoiding or escaping being with individuals around whom such experiences tend to happen [29]”. That is, avoidance of experience includes visible actions, and avoidance of thoughts and feelings in the mind. This difference in structure may be due to the difference between parents with children and ones with infants and toddlers, or it may also be due to the difference in sample size (*n* = 154 for the original PAAQ, *n* = 2000 in our study).

### 4.2. Reliability

PAAQ-J’s internal consistency and test-retest reliability were within an acceptable range. When the α coefficient of “Unwillingness” was compared with the original PAQ, it was about the same as α = 0.65 (PAAQ) vs. 68 (PAAQ-J). In addition, with regard to “Inaction”, PAAQ-J was confirmed to have higher internal consistency, and sufficient internal consistency was confirmed (total: 0.65 [PAAQ] vs. 0.80 [PAAQ-J], 0.64 [PAAQ] vs. 84, 0.72, 0.74 [PAAQ-J subscales; Inaction-B, Inaction-C, Inaction, “Inaction”], respectively). Similarly, in the test-retest reliability, PAAQ-J had an ICC of 0.49. However, since this re-examination was carried out 14 months after the initial survey, it is difficult to determine whether this figure is appropriate. This finding is therefore important, given that the first PAAQ study did not confirm the test-retesting reliability, although it was developed using the Japanese version of AAQ (PAAQ-J) [23] and was performed in an attempt to standardize it for parents with teenage children (*n* = 47) [23].

Therefore, it can be concluded that the PAAQ-J scale is a relatively stable measure of psychological flexibility for parents raising infants and toddlers.

### 4.3. Criterion and Construct Validity

The study further noted a significant positive correlation between PAAQ-J and AAQ-II. Criterion validity was further demonstrated, as AAQ-II is a gold-standard scale for evaluating the avoidance of experiences and psychological flexibility. A significant correlation was also noted between PAAQ-J and HADS-depression and HADS-anxiety. Avoidance of experiences is related to depression and anxiety symptoms. ACT is considered effective in alleviating these symptoms [30]. Therefore, from the viewpoint of construct validity, it is generally thought that PAAQ-J measures psychological flexibility for parents raising infants and toddlers. The number of subjects for evaluating PAAQ-J was 2000, and therefore the reliability and validity were high. PAAQ-J is a 12-item, two-factor structure (“Inaction” (Inaction-B, Inaction-C), “Unwillingness”) scale, and has been determined that it has sufficient reliability and validity.

### 4.4. Limitation and Future Research Perspectives

There are several limitations to this study. First, because it is an online survey, the findings may not be representative of parents with infants and toddlers. Online surveys have the advantage of convenience of participation and the ability to target a diverse population that is not concentrated in one geographic area. Conversely, there are some disadvantages, such as the composition and representativeness of the enrollment population and the unclear relationship between the planned sample and the collection sample.

Second, it is necessary to examine the validity of the retest reliability period. Generally, it is desirable to set a period of about 3 months to confirm the ICC.

This study confirmed that PAAQ-J can appropriately evaluate experiential avoidance and psychological flexibility of individual child-rearing experiences of parents raising infants and toddlers. The number of cases of child abuse in Japan has reached a record high [1]. In addition, it is known that 77.2% of deaths due to abuse were children under 3 years of age, and 55.1% of the perpetrators were the biological mothers. The parenting stress of mothers who care for children from infancy to 3 years of age is extremely high, and parents need support. In the midst of this, attention is being paid to the support of parents using ACT. In this study, we confirmed a scale that can measure the psychological flexibility of parents with infants and toddlers, and it was a catalyst for the development of parental support for using ACT in the future. The current situation is that it is still almost impossible to receive support using ACT in Japan. In the future, it is necessary to investigate the relationship between emotional and abusive parenting and psychological flexibility, and whether interventional studies using ACT can improve parents’ stress, emotional coordination skills, and mental health such as depression.

The present study was limited to testing whether the PAAQ-J can be used for parents with infants and toddlers using external criteria. However, the significance of this study includes examining the relationship between parental psychological flexibility and other variables to identify needs for ACT intervention. We believe that future research that also focuses on internal differences will further clarify the significance of this scale.

## 5. Conclusions

This study confirmed that PAAQ-J can appropriately evaluate experiential avoidance and psychological flexibility of individual child-rearing experiences of parents raising infants to toddlers. In the future, studies are needed to examine changes in parent-child behavior, behavior, and overall mood in order to develop interventions to increase parents’ psychological flexibility.

## Figures and Tables

**Figure 1 ijerph-20-05674-f001:**
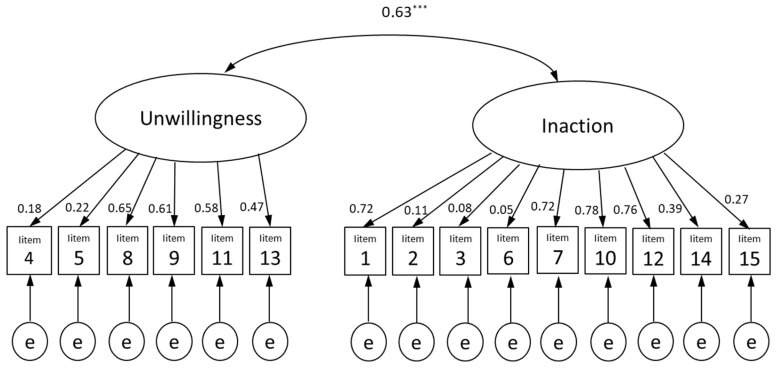
Confirmatory factor analysis of the Japanese version of the Parental Acceptance and Action Questionnaire (PAAQ-J). *** *p* < 0.001.

**Figure 2 ijerph-20-05674-f002:**
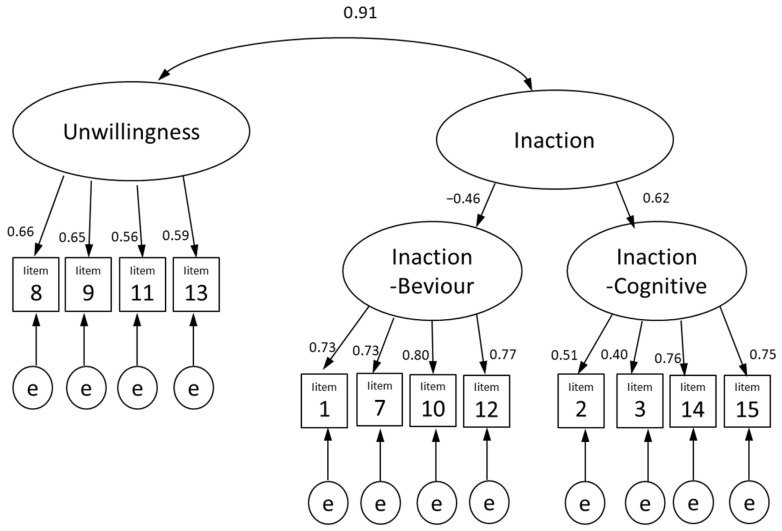
Model assuming higher order factors of the Japanese version of the Parental Acceptance and Action Questionnaire (PAAQ-J).

**Table 1 ijerph-20-05674-t001:** Sample demographic characteristics and mean (SD) of each variable.

	*n* = 2000	
	Mean	% or SD
Parent Age	33.58	4.70
Employment status		
Full-time (over 4 days)	559	28.0%
Part-time (under 3 days)	162	8.1%
Childcare leave	438	21.4%
Unemployed	829	41.4%
Number of births	2.66	0.80
Child birth order	1.51	0.74
Child Age	1.50	1.12
Current Illness		
Psychiatric disorders	50	2.5%
PMS	254	12.7%
PMDD	54	2.7%
Measures:		
Inaction-B	15.93	4.47
Inaction-C	13.62	4.45
Unwillingness	13.62	4.40
PAAQ-total	43.18	7.42
AAQ-II-total	20.12	8.35
HADS-D	8.48	3.65
HADS-A	6.01	3.79

AAQ, Acceptance and Action Questionnaire; HADS, Hospital Anxiety and Depression Scale; PAAQ, Parental Acceptance and Action Questionnaire; PMDD, premenstrual dysphoric disorder; PMS, premenstrual syndrome; SD, standard deviation.

**Table 2 ijerph-20-05674-t002:** Exploratory maximum likelihood solution analysis of PAAQ-J.

Factor	No.	Item	Factor Loadings		
			I	II	III
Inaction-B	10 *	Despite my doubts, I feel as though I can set a plan for managing my child’s feelings.	0.08	−0.11	0.78
	12 *	If I get frustrated with my child, then I can still help him or her.	0.77	−0.07	0.07
	7 *	I’m not afraid of my child’s feelings.	0.76	0.02	−0.07
	1 *	I am able to take action about my child’s fears, worries, and feelings even if I am uncertain what the right thing is to do.	0.74	0.05	−0.06
Inaction-C	15	When I compare myself to other parents, it seems that most of them are handling their lives better than I do.	0.04	0.72	−0.04
	14	I often catch myself daydreaming about things I’ve done with my child and what I would do differently next time.	0.19	0.71	−0.07
	2	When I feel depressed or anxious, I am unable to help my child manage their fears, worries, or feelings.	−0.15	0.62	−0.06
	3	I try to suppress thoughts and feelings about my child that I don’t like by just not thinking about them.	−0.18	0.46	0.15
Unwillingness	9	It is bad if my child feels anxious.	−0.09	−0.05	0.85
	11	If I could magically remove all the painful experiences my child has had in his or her life, I would do so.	0.12	0.03	0.48
	13	Worries can get in the way of my child’s success.	0.01	0.13	0.42
	8	I try hard to avoid having my child feel depressed or anxious.	0.30	0.12	0.38
		Inter-Factor Correlation	I	II	III
		I	―	−0.18	−0.04
		II		―	0.41
		III			―

PAAQ, Parental Acceptance and Action Questionnaire. * Inverted item.

**Table 3 ijerph-20-05674-t003:** Pearson’s correlation coefficient and alpha coefficient between PAAQ-J and each scale in this study (*n* = 2000).

			HADS	
	*α*	AAQ-II	Depression	Anxiety
Total	0.80	0.57 **	0.32 **	0.33 **
Inaction	0.74	0.48 **	0.36 **	0.34 **
Inaction-B	0.84	0.01	0.30 **	0.30 **
Inaction-C	0.72	0.59 **	0.19 **	0.27 **
Unwillingness	0.68	0.35 **	0.02	0.09 **

AAQ, Acceptance and Action Questionnaire; HADS, Hospital Anxiety and Depression Scale; PAAQ, Parental Acceptance and Action Questionnaire. ** *p* < 0.01.

## Data Availability

The data presented in this study are available on request from the corresponding author.

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
