# Peer review of "Psychometric Properties of the Japanese Version of the Parental Acceptance and Action Questionnaire in Parents with Infants and Toddlers"

_ijerph, 2023, doi:10.3390/ijerph20095674_

Round 1
Reviewer 1 Report
1. The study only validated the structure of the scale “ Parental Acceptance and Action Questionnaire” developed by Mizusaki and Satowith with the sample of mothers of infants aged 0–3 years and cannot be defined as the scale development. This study also did not follow the necessary steps for scale development.
2. In the introduction section, the paper's originality is not well defined: authors should underline the paper's contribution to the literature and explain the research gap that should be covered.
3. The scale validation should elaborate on its value and difference from the original scale. The authors are suggested to provide a literature review of the current state of research on the Parental Acceptance and Action Questionnaire. The insufficient sample size of the original scale alone should not be enough to support developing and validating new scales.
4. Is there a difference between the scale for mothers aged 0 to 3 years and the scale for parents with teenage children? Is it appropriate to use the original questionnaire items without any identification?
5. It is recommended that authors report the relevant results of EFA and CFA in accordance with academic norms(KMO,Bartlett's test of sphericity, χ2/df,...).
6. Authors should also compare their own results with previous findings from the literature to pinpoint the novelty of their paper in the Discussions part. Conclusions should conclude theoretical implications, managerial contributions,limitations, and future research perspectives.
Author Response
I have compiled the reconsideration into a file. Thank you in advance for your cooperation.

Reviewer 2 Report
This is a very interesting study. Helps improve the parenting style of parents.
1. The total number of documents is relatively small, which cannot reflect the relationship between this study and related research, such as innovation or breakthrough;
2. The ACT theory is the foundation of this study, but it is not further clarified in the text what kind of support this theory plays for this study.
3. "Parental Acceptance is a basic issue in this study, but this concept and related research are not introduced in the article.".
4. The data still needs to be further mined, allowing in-depth analysis of potential internal differences in a certain aspect.
Author Response

(The authors gave the same response as above.)

Round 2
Reviewer 1 Report
After revising, I think the manuscript has been improved to be considered for publication in IJERPH.